# PeerJ

# A holistic evolutionary and structural study of *flaviviridae* provides insights into the function and inhibition of HCV helicase

Dimitrios Vlachakis, Vassiliki Lila Koumandou and Sophia Kossida

Bioinformatics & Medical Informatics Team, Biomedical Research Foundation, Academy of Athens, Athens, Greece

## ABSTRACT

Viral RNA helicases are involved in duplex unwinding during the RNA replication of the virus. It is suggested that these helicases represent very promising antiviral targets. Viruses of the *flaviviridae* family are the causative agents of many common and devastating diseases, including hepatitis, yellow fever and dengue fever. As there is currently no available anti-*Flaviviridae* therapy, there is urgent need for the development of efficient anti-viral pharmaceutical strategies. Herein, we report the complete phylogenetic analysis across *flaviviridae* alongside a more in-depth evolutionary study that revealed a series of conserved and invariant amino acids that are predicted to be key to the function of the helicase. Structural molecular modelling analysis revealed the strategic significance of these residues based on their relative positioning on the 3D structures of the helicase enzymes, which may be used as pharmacological targets. We previously reported a novel series of highly potent HCV helicase inhibitors, and we now re-assess their antiviral potential using the 3D structural model of the invariant helicase residues. It was found that the most active compound of the series, compound C4, exhibited an $IC_{50}$ in the submicromolar range, whereas its stereoisomer (compound C12) was completely inactive. Useful insights were obtained from molecular modelling and conformational search studies via molecular dynamics simulations. C12 tends to bend and lock in an almost "U" shape conformation, failing to establish vital interactions with the active site of HCV. On the contrary, C4 spends most of its conformational time in a straight, more rigid formation that allows it to successfully block the passage of the oligonucleotide in the ssRNA channel of the HCV helicase. This study paves the way and provides the necessary framework for the in-depth analysis required to enable the future design of new and potent anti-viral agents.

Corresponding author
Sophia Kossida,
skossida@bioacademy.gr

## INTRODUCTION

Viruses of the *flaviviridae* family infect vertebrates and they are primarily transmitted through arthropod vectors (mainly ticks and mosquitoes) (*Neyts, Leyssen & De Clercq, 1999*). The *flaviviridae* family includes four genera: *Flavivirus* (Yellow fever virus, West

Nile virus, Dengue virus, Tick-borne encephalitis viruses), *Hepacivirus* (Hepatitis C virus), *Pestivirus* (Classical swine fever virus, Bovine viral diarrhea virus) and Unclassified *Flaviviridae* (Hepatitis GB virus, GB viruses) (*Shepard, Finelli & Alter, 2005*). *Flaviviridae* have monopartite, linear, single-stranded, positive sense RNA genomes (*Vlachakis, 2009*), ranging from 10 to 12.5 kilobases (kb) in length (*Poynard, Bedossa & Opolon, 1997*). Because the viral RNA has positive sense, the nucleic acid itself is capable of causing an infection in the appropriate host cells. The 3'-termini of their RNA are not polyadenylated. The 5'-termini of the viral RNA genome in members of the *Flavivirus* genus have a methylated nucleotide cap, which allows translation. Sometimes it is possible to have a genome-linked protein (VPg) in place of the methylated nucleotide cap. In members of the *Pestivirus* and *Hepacivirus* genera, the 5' ends are uncapped and have an internal ribosome entry site (IRES) instead (*Nulf & Corey, 2004*), which is responsible for providing a site for the initiation of the translation process for host ribosomes (*Daly & Ward, 2003*). In both genera, structural genes are located towards the 5' end of the RNA.

Virions of the *flaviviridae* family are enveloped and slightly pleomorphic during their life cycle. They are spherical in shape and usually 40–60 nm in diameter. Their nucleocapsids are isometric and sometimes penetrated by stain. The usual size of the nucleocapsids is 25–30 nm in diameter and they have polyhedral symmetry (*Guzmán & Kourí, 2004*). It is remarkable that *flaviviridae* viruses manage to retain high similarity in the morphology of the virion, the organization of the viral genome, and the estimated life cycles and replication patterns that they follow, even though the different genera do not have common biological properties and do not show serological cross-reactivity (*Thompson & Finch, 2005*; *Rosenberg, 2001*).

One of the most important pathogens of the *flaviviridae* viral family is the Hepatitis C virus (HCV). Recent classification analysis suggests that there are 6 different genotypes of the naturally occurring virus. Approximately 180 million people are chronically infected with HCV, according to the World Health Organization. In addition to that, an estimated 4 million new infections occur every year (*Calisher & Gould, 2003*; *Kadare & Haenni, 1997*). The progression of the disease upon infection is rapid, as in more than 80% of patients a persistent infection will be established, which will eventually progress to liver cirrhosis and in most cases, hepatocellular carcinoma. The fatality rates of HCV-related diseases are estimated to be approximately 400,000 patients per year. To date no effective drug or vaccination agent is available. More importantly, the molecular machinery that the virus employs in its lethal quest is still unclear.

Inhibition of the viral helicase is an emerging and very promising approach that is becoming increasingly popular (*Phoon et al., 2001*). Helicases are capable of unwinding double stranded DNA and RNA to single strands by breaking the series of hydrogen bonds that keep the two strands together. The unwinding activity of the viral helicase is essential to the virus during its replication process. Mutated inactive helicases in Dengue and Bovine Diarrhea viruses led to reduced proliferation of the virus (*Diana & Bailey, 1997*). Therefore, it is considered that the effective inhibition of the viral helicase will be an operative tool for the suppression of the replication rate of *flaviviridae* viruses

proliferation. The viral Helicase is coded in the NS3B region of the viral genome next to the NS3A gene, which codes for the viral protease.

Many drug-like potent inhibitors of HCV have been reported in the literature, but only one that is capable of covalently interacting with the HCV helicase enzyme (*Kandil et al., 2009*). In this study, we present an in-depth updated phylogenetic analysis of *flaviviridae* helicase protein sequences. Collectively, 87 complete polyprotein sequences were retrieved from the Virus Pathogen Resource (ViPR) Database, the distribution of which in the *flaviviridae* genera is: 69 *flavivirus*, 11 *hepacivirus*, 7 *pestivirus* (*Vlachakis, 2009*; *Brancale, Vlachaki & Vlachakis, 2008*; *Vlachakis & Brancale, 2007*). The multiple sequence alignments of the NS3 protein from phylogenetically distinct species, reveal a large number of conserved residues, which had not been studied previously. As sequence conservation denotes functional importance, this result allowed for a more focused and comprehensive molecular modelling approach of HCV, which aims to provide invaluable insights that will rationalize the findings of our previously published work (*Kandil et al., 2009*) and deliver the means required for the future de novo structure-based drug design of potent anti-HCV agents.

## METHODOLOGY

### Database sequence search

Sequence data from species for which full genome data is available were obtained from the NIAID Virus Pathogen Database and Analysis Resource (ViPR) online through the web site at http://www.viprbrc.org (*Pickett et al., 2012*), as well as the NCBI RefSeq database. This selection was checked against, and supplemented with, data from previous publications regarding *flaviviridae* phylogeny (*Cook & Holmes, 2006*; *Ferron et al., 2005*; *Lobo et al., 2009*). Multiple HCV genotypes were included, but otherwise, for each species, only one strain was analysed. In total, sequences from 7 *pestivirus*, 11 *hepacivirus* (including the 6 HCV genotypes), and 69 *flavivirus* species were used, as shown in Table S1. About half of these species have fully annotated genomes; in these cases the annotated NS3 sequence was used (Table S1). For the rest, the start and end positions of the NS3 sequence within the whole genome polyprotein was inferred from alignments with closely related annotated species. Thus, only the amino acid sequence corresponding to NS3, determined in this way, was used for the multiple sequence alignments and phylogenetic analysis (for details see Table S1).

### Alignments and phylogenetic analysis

Alignments of the NS3 protein sequence (File S1) were created using MUSCLE, version 3.7 (*Edgar, 2004*), and checked visually using Jalview, version 2.7 (*Waterhouse et al., 2009*). Only unambiguous homologous regions were retained for phylogenetic analysis; manual masking and trimming was performed in MacClade, version 4.08 (*Maddison & Maddison, 1989*). ProtTest, version 1.3 (*Abascal, Zardoya & Posada, 2005*) was used to estimate the appropriate model of sequence evolution. Phylogenetic analysis was performed by three separate methods. To obtain the Bayesian tree topology and posterior probability values,

the program MrBayes, version 3.1.2 was used (*Ronquist & Huelsenbeck, 2003*). Analyses were run for 1 million generations, removing all trees before a plateau established by graphical estimation. All calculations were checked for convergence and had a splits frequency of <0.1. Maximum-likelihood (ML) analysis was performed using PhyML, version 3.0 (*Guindon & Gascuel, 2003*) and RAxML, version 7.0.0 (*Stamatakis, 2006*) with 100 bootstrap replicates. Trees were visualised using FigTree v1.2 (http://tree.bio.ed.ac.uk/software/figtree/). Nodes with at least 0.95 posterior probability and 80% bootstrap support were considered robust, and nodes with at least 0.80 posterior probability and 50% bootstrap support are indicated.

## Molecular docking

In order to elucidate *in silico* the 3D structures of compounds C4, C12 and the HCV helicase the docking suite ZDOCK, version 3.0 was used (*Chen, Li & Weng, 2003*). ZDOCK is a protein-protein docking suite that utilizes a grid-based representation of the molecular system involved. In order to efficiently explore the search space and docking positions of the molecules as rigid bodies, ZDOCK takes full advantage of a three-dimensional fast Fourier transformation algorithm. It uses a scoring function that returns electrostatic, hydrophobic and desolvation energies as well as performing a fast pairwise shape complementarity evaluation. Moreover, it uses the contact propensities of transient complexes of proteins to perform an evaluation of a pairwise atomic statistical potential for the docking molecular system. RDOCK was utilized to refine and quickly evaluate the results obtained by ZDOCK ZDOCK (*Li, Chen & Weng, 2003*). RDOCK performs a fast minimization step to the ZDOCK molecular complex outputs and ranks them according to their re-calculated binding free energies.

## Energy minimizations

Energy minimization was performed initially to remove the geometrical strain from the top-ranking hits of the docking experiments, since the proteins were treated as rigid bodies. Protein complexes were subjected to an extensive energy minimization run using the Amber99 (*Duan et al., 2003*) forcefield as it is implemented into the Gromacs, version 4.5.5 (*Hess et al., 2008*), via the Gromita graphical interface, version 1.07 (*Sellis, Vlachakis & Vlassi, 2009*). An implicit Generalized Born (GB) solvation was chosen at this stage, in an attempt to speed up the energy minimization process.

## Molecular dynamics

In order to further explore the interaction space and binding potential of each docking conformation, the molecular complexes were subjected to unrestrained molecular dynamics simulations using the Gromacs suite, version 4.5.5 (*Hess et al., 2008*). Molecular dynamics took place in a periodic environment, which was subsequently solvated with SPC water using the truncated octahedron box extending to 7Å from each molecule. Partial charges were applied and the molecular systems were neutralized with counter-ions as required. The temperature was set to 300 K and the step size was set to 1 fs. The total run of each molecular complex was fifty nanoseconds, using the NVT ensemble in a canonical

environment. NVT stands for Number of atoms, Volume and Temperature that remain constant throughout the calculation. The results of the molecular dynamics simulations were collected into a molecular trajectory database for further analysis.

## Molecular dynamics analysis

Principal component analysis was done using Pymol, version r0.99 (*DeLano, 2002*) and the Ca atom root-mean-square function of Deep-View, version 4.0.1 (*Guex & Peitsch, 1997*). Analysis of the Molecular Dynamics outputs and trajectories was therefore focused on structural deviations of each molecular system from its original docking conformation. The molecular dynamics final conformations were initially evaluated with a residue packing quality function built-in the Gromacs suite, which depends on the number of buried non-polar side chain groups and on hydrogen bonding. Moreover, the suites Procheck, version 3.5 (*Laskowski et al., 1996*) Verify3D, version 1.0 (*Eisenberg, Luthy & Bowie, 1997*) were employed to evaluate the structural viability of each protein complex upon the molecular dynamics simulations. Illustrations of the molecular systems were rendered with the aid of the Chimera suite, version 1.8 (*Pettersen et al., 2004*).

# RESULTS/DISCUSSION

## Alignments and phylogenetic analysis

The alignment of *Hepacivirus* NS3 protein sequences (Fig. S1), across 11 different species, shows extremely high conservation throughout the whole length of the sequence, including 119 residues which are absolutely conserved among all the species examined. This indicates a strong selection pressure to prevent sequence change and may highlight important functional domains of *Hepacivirus* NS3, in addition to those that have been identified previously.

The alignment of NS3 protein sequences across 87 *flaviviridae* species (Fig. S2) shows good conservation throughout the whole length of the sequence, particularly between species that belong to the same genus (i.e. *Hepacivirus*, *Flavivirus*, *Pestivirus*). The annotated *Pestivirus* NS3 sequences (∼1136 aa) are significantly longer than the annotated NS3 sequences from *Flavivirus* (∼619 aa) and *Hepacivirus* (∼631 aa). *Pestivirus* NS3 shows extensive insertions in the N-terminal half, compared to *Flavivirus* and *Hepacivirus* NS3. Regions conserved across all three genera likely indicate important functional domains of the NS3 protein, and the alignment highlights 24 residues which are absolutely conserved between all species. Many of the conserved regions identified here have not been reported previously, and probably deserve further study.

Phylogenetic reconstruction of the NS3 protein sequences from all 87 *flaviviridae* species (Fig. 1) shows clear separation of the *Pestivirus*, *Hepacivirus*, and *Flavivirus* species. Within the *Flavivirus*, monophylletic groups for the tick-borne and insect-specific *Flavivirus* are evident. The insect-specific *Flavivirus*, as well as TABV appear basal to (closest to the origin of) the whole *Flavivirus* group. Mosquito-borne species diverge afterwards, but before tick-borne species, and species with no known vector (NKV). This is largely in agreement with previous analyses based on NS3 sequence data, with some

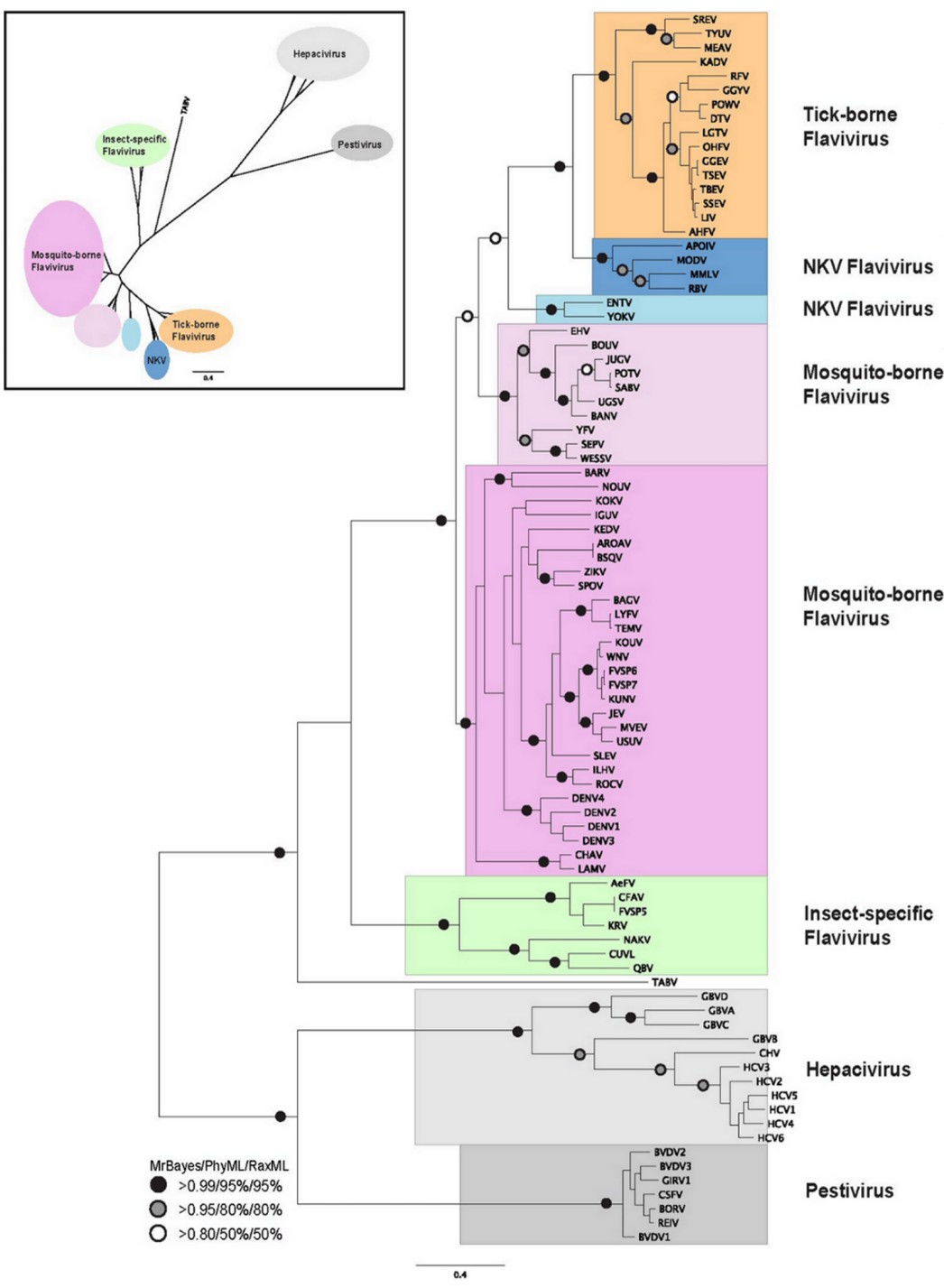

**Figure 1 Phylogenetic reconstruction of *flaviviridae* NS3 protein sequences.** The tree shown is the best Bayesian topology. Numerical values at the nodes of the tree ($x/y/z$) indicate statistical support by MrBayes, PhyML and RAxML (posterior probability, bootstrap and bootstrap, respectively). Values for highly supported nodes have been replaced by symbols, as indicated. Full details and accession numbers (continued on next page...)

**Figure 1 (...continued)**
for all protein sequences used are given in Table S1. The tree confidently separates the *Hepacivirs*, *Pestivirus*, and *Flavivirus* genera. Within the *Flavivirus*, TABV and insect-specific species appear basal, whereas Tick-borne species and species with no known vector (NKV) are the most derived. The inset shows the same tree in unrooted star format to better illustrate the relationships and distances between the subgroups.

variations on the order of events probably resulting from the use of a greater number of species in our analysis than has been used previously, as well as a more confident placement of the root of the *Flavivirus* genus, due to the inclusion of a number of *Pestivirus* and *Hepacivirus* sequences. Most of the groupings in the tree are highly supported by posterior probability and bootstrap values by all three phylogenetic methods (Fig. 1).

Phylogenetic reconstruction of the Hepacivirus NS3 protein sequences (Fig. 2) indicates two subgroups, one consisting of the six HCV genotypes, and the other consisting of GBVA, GBVC, GBVD. GBVB and CHV are between these two subgroups, with GBVB grouping closer to the other GBVs, and CHV closer to the HCVs. The phylogenetic analysis demonstrates the evolutionary distances of *flaviviridae* and *Hepacivirus* species used in this study, lending support to the conclusions we draw about sequence conservation relating to function (see below).

## Evolutionary study/invariant residues

Further, protein motifs were derived from the multiple alignments of the *Flaviviridae* helicase amino acid sequences (Fig. 3). Apart from the conserved and already reported sequence motifs that are characteristic to the family of *flaviviridae* (*Vlachakis, 2009*), a series of conserved/invariant residues were also identified from our multiple alignments. This is a finding of great importance as it demonstrates the importance and significance of various amino acids that remain invariant, even though they do not participate in the known conserved motifs of the helicase enzymes in the *Flaviviridae* family of viruses. Residue conservation across the diverse *flaviviridae* family is indicative of the importance and necessity of these residues to the function of the viral helicase. Working on the principle that in nature structure is more conserved than sequence and therefore more closely related to biological function, an effort was made to map the position of the identified invariant amino acids on the 3D structure of a set of representative helicase enzymes from the *flaviviridae* family. In this direction the crystal structures of the helicase enzymes of HCV (RCSB PDB code: 1A1V), Yellow Fever virus (RCSB PDB code: 1YKS), Dengue virus (RCSB PDB code: 2BMF) and Murray Encephalitis virus (RCSB PDB code: 2V8O) were structurally superimposed. The identified conserved amino acids were selected and are shown in spacefill representation in Fig. 4. The co-crystallised ssRNA fragment as well as the Magnesium atom that defines the ATP hydrolysis site (*Kim et al., 1998*) have been borrowed from the HCV helicase structure and are also shown in Fig. 4 in spacefill green color representation. Figure 4, displays the conserved residues in spacefill CPK representation for four different helicase X-ray structures: HCV, Dengue, Yellow Fever and Murray Valley Encephalitis helicases.

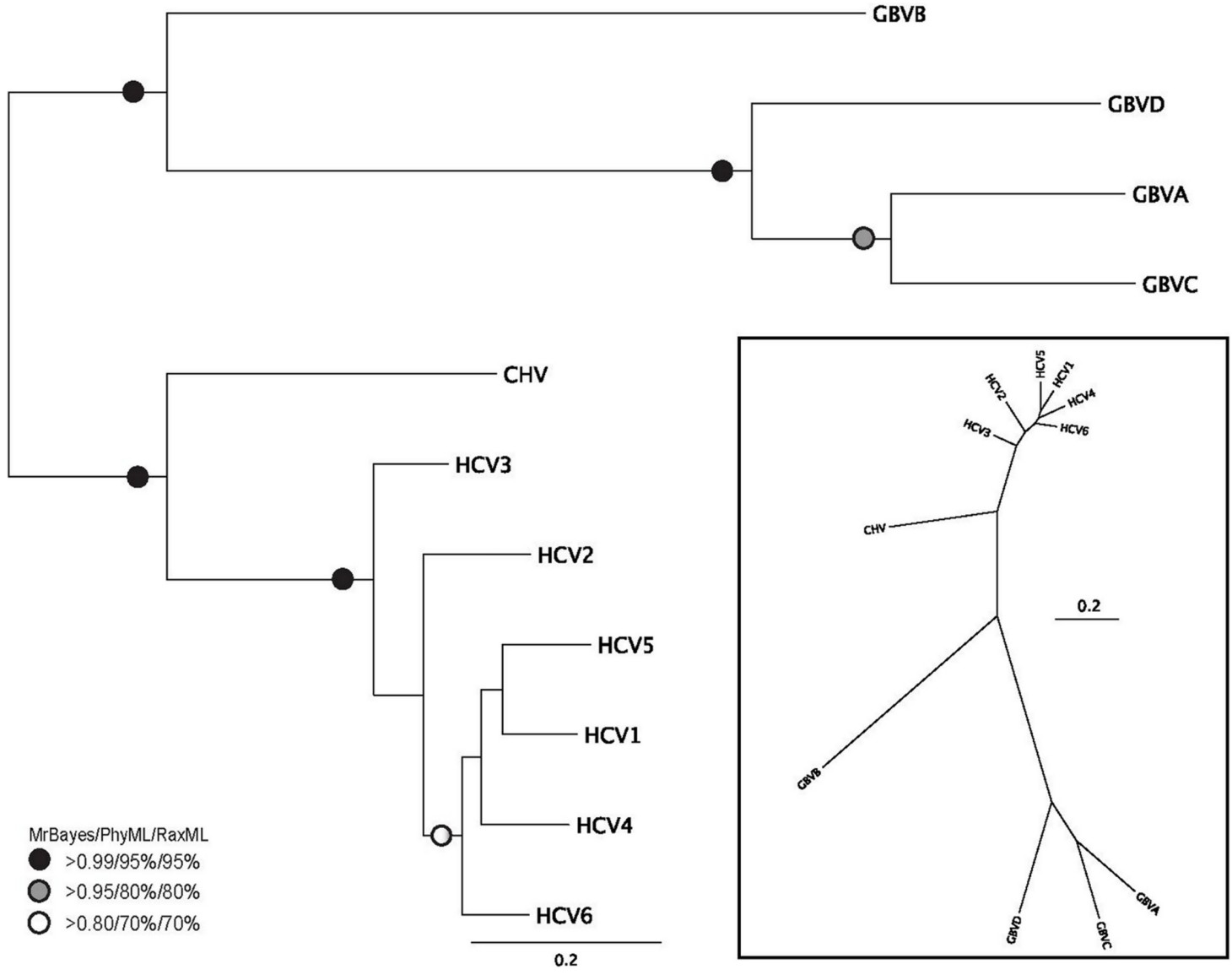

**Figure 2** **Phylogenetic reconstruction of *Hepacivirus* NS3 protein sequences.** The tree shown is the best Bayesian topology. Numerical values at the nodes of the tree ($x/y/z$) indicate statistical support by MrBayes, PhyML and RAxML (posterior probability, bootstrap and bootstrap, respectively). Values for highly supported nodes have been replaced by symbols, as indicated. Full details and accession numbers for all protein sequences used are given in Table S1. The tree indicates two subgroups, one consisting of GBVA, GBVB, GBVC, and GBVD, and the other consisting of CHV and the six HCV genotypes. The inset shows the same tree in unrooted star format to better illustrate the relationship and distance between the subgroups.

Strikingly, it was found that the conserved amino acids from our phylogenetic analysis are not randomly distributed on the helicase structure. On the contrary, they are strategically positioned in "key" regions with distinct enzymatic activity. The latter 3D conformational positions provide invaluable information to the medicinal chemistry antiviral domain, as they constitute pharmacological targets for the designing of new potent antiviral agents. Overall, three separate regions were identified (Fig. 4). The first region of the helicase enzymes of *flaviviridae* is the ATP hydrolysis site. A well organised,

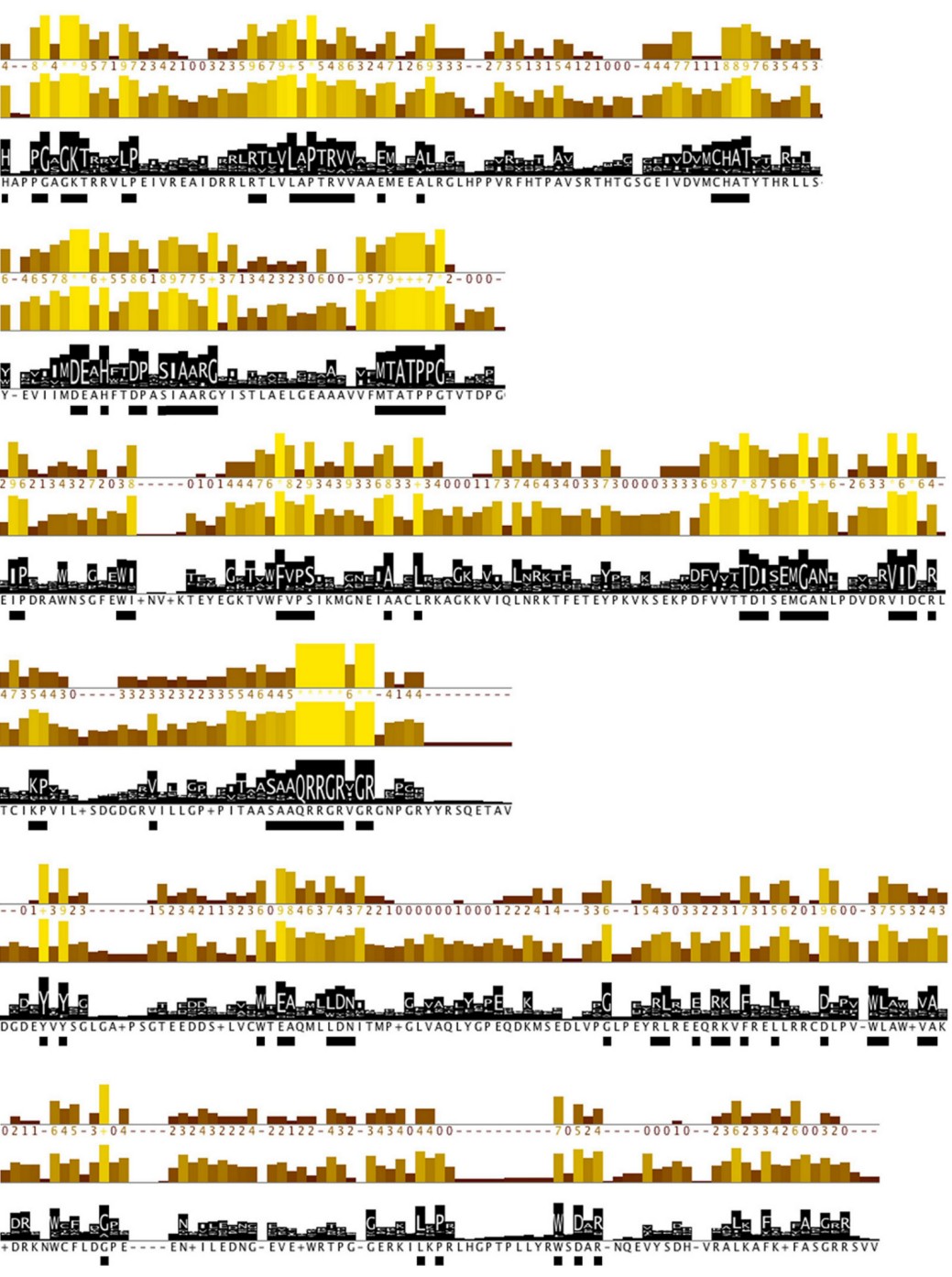

**Figure 3 Conserved residues identified by analysis of all *flaviviridae* NS3 proteins.** Conservation, quality and consensus tracks from Jalview for selected regions of the alignment shown in Fig. S2 are shown. Regions of high conservation are underlined with black boxes. The conservation annotation histogram (top) reflects conservation of the physicochemical properties, and marks absolutely conserved residues (score 11) with a yellow asterisk '*', and columns where physicochemical properties are conserved (score 10) with a yellow '+'; less conserved positions are shown in darker colours with decreasing score. The quality annotation histogram (middle) (continued on next page...)

**Figure 3 (...continued)**
reflects the likelihood of observing a mutation in any particular column of the alignment based on the BLOSUM62 matrix scores (for each column, the sum of the ratios of the two BLOSUM62 scores for a mutation pair, and each residue's conserved BLOSUM62 score, are normalised and plotted on a scale of 0 to 1). The consensus histogram (bottom) reflects the percentage of the modal residue per column, and the consensus sequence logo is shown for conserved regions ('+' denotes non-conserved residues and '-' denotes gap residues).

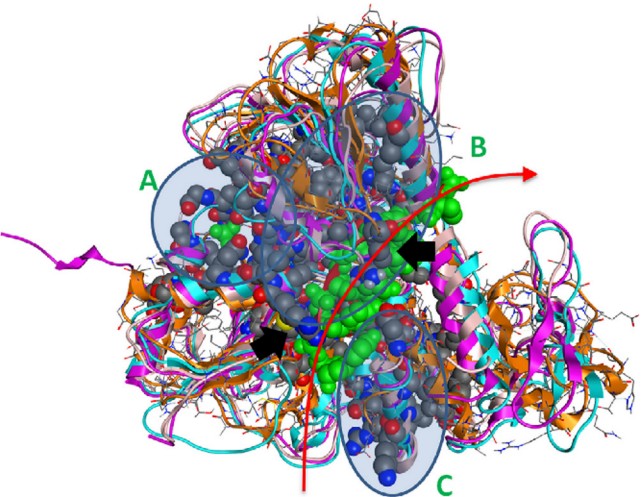

**Figure 4 The 3D structure of HCV helicase showing all conserved amino acid groups highlighted in Fig. 3 in all four helicase X-ray structures.** The two arginine residues are indicated by black arrows. The route that the ssRNA fragment follows through the helicase is shown by the red arrow. A, B and C are the three regions of sequence and structural conservation. Four X-ray determined helicase structures have been superposed here. HCV helicase is orange, Dengue helicase is magenta, Yellow fever helicase is pink and Murray Valley Encephalitis helicase is turquoise.

pocket-like active site is established by invariant residues across the *flaviviridae* species. There, a set of invariant residues (including the regulating aspartic acid amino acid) coordinate the hydrolysis of the ATP substrate by optimal coordination of the 2 $Mg^{++}$ atoms. The second conserved region is the core of the third domain of the helicase enzymes. There, a set of antiparallel beta-sheets organize a unique three dimensional arrangement that is structurally quite robust. The stiffness of this domain is very important and directly linked to the functionality of the helicase. Remarkably, the main ssRNA interacting residues from both A and B regions are two arginine amino acids. Those positively charged residues are expected to interact with the negatively charged backbone of the oligonucleotide substrate (Fig. 4, black arrows). Similarly, the third region that is comprised of conserved residues is located in-between the 1st and 2nd domains of the helicase protein. This is believed to be the entrance site for the incoming oligonucleotide during the dsRNA unwinding. In region C, we observe a lysine and arginine rich organization of conserved residues. The cumulative positive charge of the latter is linked to the welcoming, electrostatic forces that initiate the processing of the oligonucleotide chain by the viral helicase enzyme. Overall, the combination of the newly identified invariant

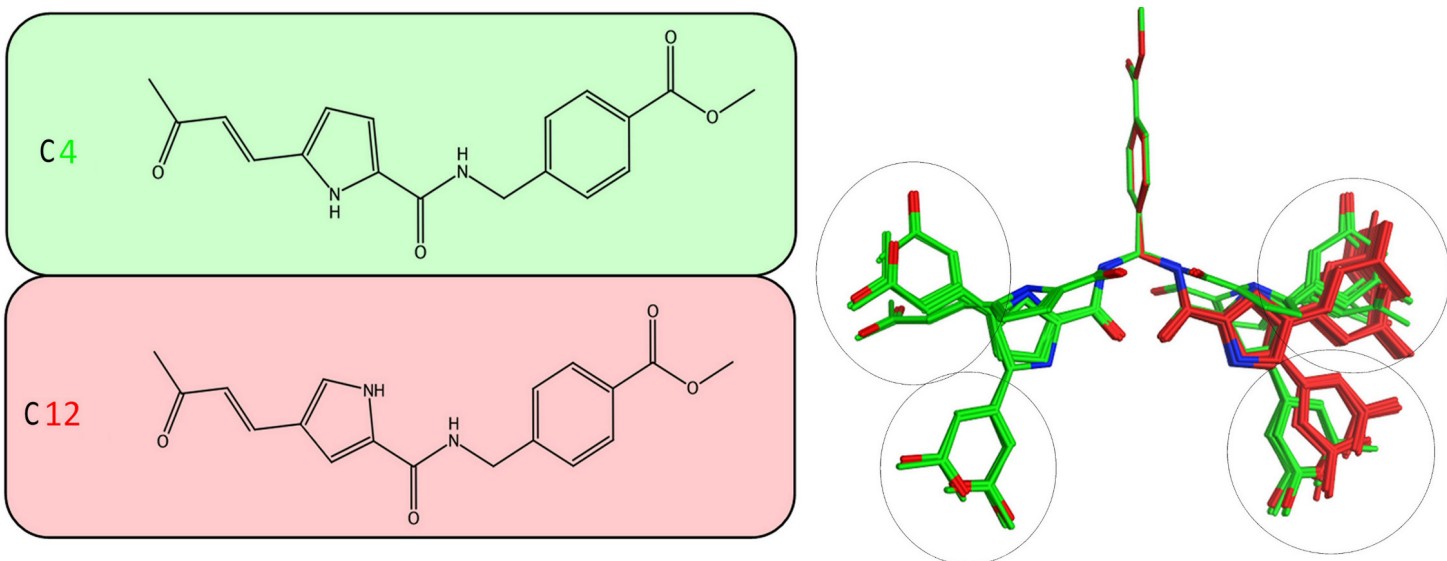

**Figure 5 Stochastic conformational search for C12 and C4.** It was established that C12 only formed two relatively close clusters (coloured red), whereas C4 formed two sets of conformationally distant ones (coloured green).

residues, the charge distribution on the 3D structure of the HCV helicase and the relative spatial arrangement of the helicase motifs constitute irreplaceable information than may lead to the better understanding of the mechanism of action of the HCV helicase and the consequent development of more potent anti-viral agents.

## Structural study of the HCV helicase

Herein, an effort is made to combine all of the above evolution-based findings with the available 3D structural information of *flaviviridae* helicases in order to explain and rationalize the assay behaviour of our recently published HCV helicase inhibitors (*Kandil et al., 2009*). The key element in this series of compounds (Fig. 5) is the Michael acceptor moiety that they bear, which is capable of establishing strong covalent bonds with a cysteine residue (Cys431), located in the ssRNA channel of the HCV helicase. As soon as the Cys431 interaction is established, another hydrogen bond is formed with Arg481, which is adjacent to the Cys431 residue (Figs. 6B and 6C). A further hydrogen bond to Val432 helps C4, but not C12, to acquire the optimal directional axis required in the ssRNA channel, which is in accordance with our previously proposed mode of helicase inhibition for compounds C4 and C12. This is achieved by establishing H-bonding interactions with the solvent-exposed Arg393 amino acid. Therefore, potent compounds of this series should be able to not only establish covalent interactions with Cys431 and hydrogen bonding interactions with Arg481 but also be able to retain a conformationally rigid pose and reach the Arg393 residue. Hydrogen bonding to Arg393 would fixate the potential of those compounds in the three dimensional space of the ssRNA HCV channel, thus inhibiting the enzyme.

It was determined by our in-house, custom developed enzymatic assay (*Kandil et al., 2009*) that whilst C4 is active in the submicromolar range, its stereoisomer counterpart

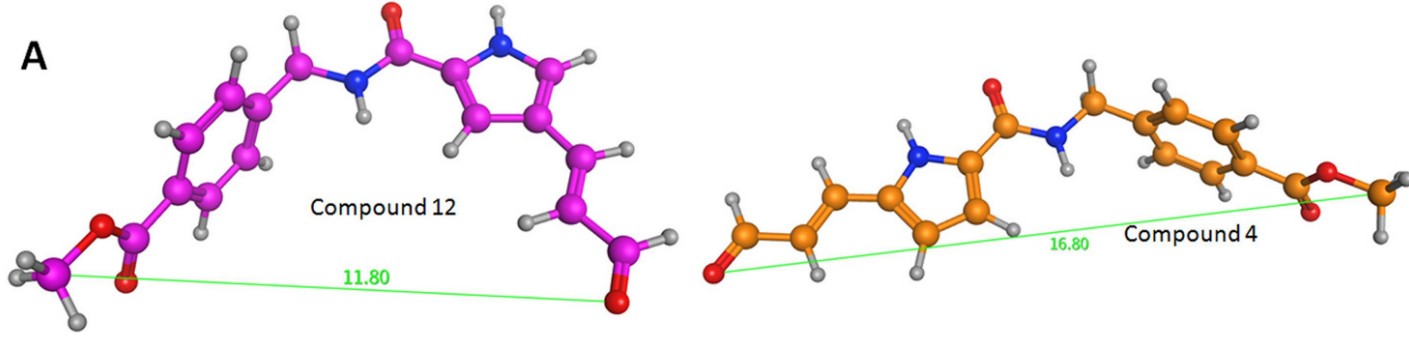

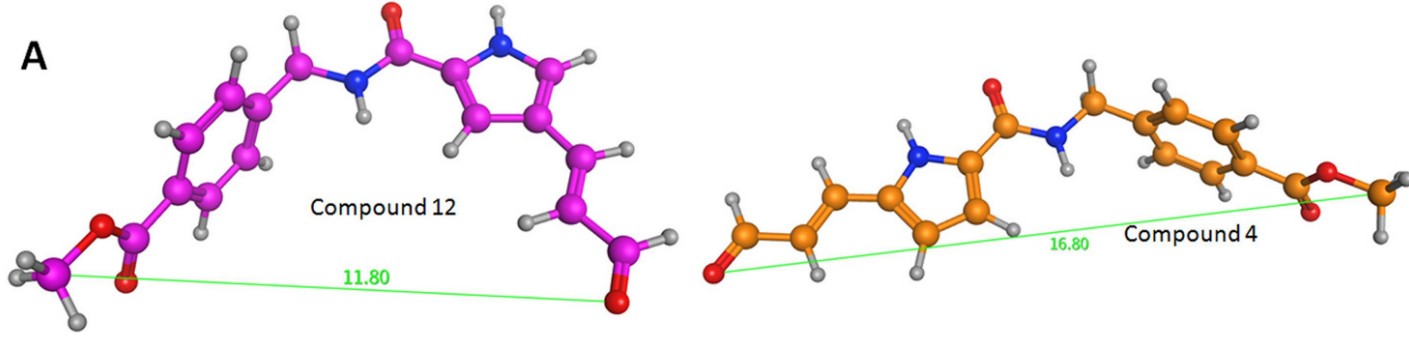

**Figure 6 Conformational analysis for compounds C4 and C12.** (A) The most prominent conformations adopted by C4 and C12. (B) C4 in the ssRNA (blue ribbon) of HCV helicase. (C, Upper) The interaction of C4 with both arginine residues of the ssRNA channel is blocking the passage of the oligonucleotide. (C, Lower) Having established the Cys431 bond, C12 fails to reach Arg393 and form a blocking bridge in the ssRNA channel of the HCV helicase. Compounds have been color-coded as in A. Electrostatic potential surfaces for C4 have been drawn in B and C.

is completely inactive. That was very strange as some activity was expected from C12 too. Since the only difference between C4 and C12 is their unique stereochemical variation, our molecular modelling approach allows us to correlate it with their observed inhibitory potentials. Initially a stochastic conformational analysis was executed *in vacuo* for compounds C4 and C12 using the implemented module in the Molecular Operations Environment suite (MOE-2011.10). The degrees of freedom available for each compound can be quantified and visualised, in order to identify the amount and distribution of the 3D conformational space that each compound can occupy. To perform a direct comparison between the two stereoisomers, the two counterpart benzene-COOH moieties were structurally superposed and their potential in the 3D space was fixed. The stochastic conformational search simulation was run for a total of 10000 iteration steps. Analysis of the simulation revealed that compound C4 explored approximately 30% more conformational space than compound C12. Namely, C4 explored 400 cubic Angstroms (c$\text{Å}^2$), while C12 explored only 270 c$\text{Å}^2$. Moreover, while C12 seems to be trapped between two relatively close conformational clusters, C4 explored the conformational space in all directions between two separate conformational clusters, each one of which consisted of two relatively close sub-clusters (Fig. 5). However, a more interesting observation was that C12 spent most of the simulation time moving from an extended to a bended "U" like shaped conformation (Fig. 6A). That led to a series of extensive docking and molecular dynamics simulations, in an effort to investigate the conformational preferences of each stereoisomer in the ssRNA channel of the HCV helicase.

The molecular docking experiments were performed using two distinctly different algorithms: the MOE docking module and ZDock. It is of great interest to determine which one is best for docking small molecule drug-like compounds in large docking sites, such as the helicase ssRNA channel. Surprisingly, they produced very similar results, even though they are based on different principles. MOE dock uses an Alpha Triangle displacement method, suitable for rotating bonds and flexing small molecule compounds in an active site. On the other hand, Zdock is a docking suite that utilizes a grid-based representation of the molecular system involved. In order to efficiently explore the search space and docking positions of the molecules as rigid bodies, ZDock takes full advantage of a three-dimensional fast Fourier transformation algorithm. It uses a scoring function that returns electrostatic, hydrophobic and desolvation energies as well as performing a fast pairwise shape complementarity evaluation. A set of 100 top ranking poses for C4 and another identical set of poses for C12 from the previously performed conformational search were used as input structures for Zdock. For MOE-Dock an ideal energetically minimised structure of each of the two stereoisomers was used, while the number of output poses was limited to 100 per compound. That is because MOE-Dock can generate conformations from a single 3D conformation by applying a collection of preferred torsion angles to the rotatable bonds. MOE-Dock uses built-in scoring functions, where lower scores indicate more favorable poses. The units for all scoring functions are kcal/mol. On the other hand, the RDOCK algorithm must be utilized to refine, score and evaluate the results obtained

by ZDOCK. RDOCK performs a fast minimization step to the ZDOCK molecular complex outputs and ranks them according to their re-calculated binding free energies.

Following the docking experiments a series of molecular dynamics simulations were initiated for the two stereoisomers in the active site of the HCV helicase. The molecular docking results were filtered and only those interacting with the previously reported key residues of the HCV helicase channel were selected. Molecular complexes consisting of the HCV helicase receptor and the highest ranking docking poses for C4 and C12 were subjected to unrestrained molecular dynamics simulations using the MOE molecular dynamics module. The Michael acceptor covalent bond between each inhibitor compound and the receptor was manually established to ensure that the compounds would remain in the active site's proximity. All molecular complexes were energetically minimized prior to molecular dynamics, in order to relax any potential geometrical strain in the starting files. Molecular dynamics simulations were performed in a periodic box that was subsequently solvated with simple point charge (SPC) water. Partial charges were applied using the built-in MOE PEOE utility. Counter ions were added accordingly as required to neutralize the molecular system. Temperature was set to 300 K, pressure to 1 atm and the step size to 1 fs. The total run of each helicase – inhibitor molecular complex was 50 ns, using the default NVT ensemble in a canonical environment. NVT simulations retain the number of atoms, Volume and Temperature constant throughout the calculations. Analysis of the molecular dynamics simulation trajectories (Figs. 6B and 6C) revealed that C12 tends to bend, forming a 45 to 75 degree angle that gives it a "U" like shape and makes it less rigid while, for the majority of its conformational time, the compound remains shorter than C4 at an average length of 11.8 angstroms. C4's average length throughout the simulation was measured to be 16,8 angstroms (Fig. 6A). C4 is far more rigid and ideal for the formation of the necessary hydrogen bonds with the two arginine residues in HCV helicase.

Based on the biological data previously published on the two stereoisomers and the current 3D modelling study, we conclude that the compound's steric rigidity, which locks it in a 3D linear axis conformation is essential for the activity of this family of compounds. Moreover, a speculation is made on the potential activity of C4 against the Dengue helicase, as it was observed that the ssRNA channel of Dengue helicase overlaps with high accuracy with its corresponding channel on the HCV helicase (data not shown). More interestingly, it was found that the cysteine residue of HCV (Cys431) superposes with another cysteine that is located in the same position of the Dengue helicase channel, when the two enzymes are structurally superimposed. Therefore the latter cysteine residues are key candidates to be considered as pharmacological targets, as they are both exposed to solvent, while strategically located in the centre of the ssRNA channel of the viral helicase. Taken together, it was demonstrated that C4 adopts a rather linear-extended conformation throughout the molecular dynamics simulations. This property allows it to hydrogen bond to Arg481, Val432 and Arg393 while being covalently bound to Cys431. Collectively, it is proposed that the hydrogen bonding to Arg393, Arg481, Val432 and the covalent bond to Cys431 are responsible for fixating C4 in the ssRNA channel of HCV, which irreversibly blocks the passage of the incoming oligonucleotide and potently inhibits the HCV helicase enzyme.

Modern computer-based methodologies are now becoming an integral part of the drug discovery process that may eventually lead to the development of promising and effective antiviral strategies. Overall, the current systematic and thorough study of the HCV helicase in terms of its unique structural characteristics, the underlying mechanism that is responsible for the difference observed in the activity of the C12 and C4 stereoisomers and the information obtained by our in-depth phylogenetic and evolutionary study, provide invaluable insights into the mechanism of action of this viral enzyme and may be used for the design of novel compounds that will effectively inhibit the action of the helicase, subsequently stopping the proliferation and infection of the virus.

## ACKNOWLEDGEMENTS

The authors would like to sincerely thank Spyridon Champeri Tsanira for critically reviewing the manuscript. The authors also acknowledge the contribution of data deposited in the ViPR database, to the present study. The Virus Pathogen Database and Analysis Resource (ViPR) has been wholly funded with federal funds from the National Institute of Allergy and Infectious Diseases, National Institutes of Health, Department of Health and Human Services, under Contract No. HHSN272200900041C.

### Funding

VLK is funded through an FP7-PEOPLE-2011-IEF Marie Curie fellowship. The funders had no role in study design, data collection and analysis, decision to publish, or preparation of the manuscript.

### Grant Disclosures

The following grant information was disclosed by the authors:
FP7-PEOPLE-2011-IEF Marie Curie fellowship.

### Competing Interests

The authors declare there are no competing interests.

### Author Contributions

- Dimitrios Vlachakis and Sophia Kossida conceived and designed the experiments, performed the experiments, analyzed the data, contributed reagents/materials/analysis tools, wrote the paper.
- Vassiliki Lila Koumandou performed the experiments, analyzed the data, contributed reagents/materials/analysis tools, wrote the paper.

### Patent Disclosures

The following patent dependencies were disclosed by the authors:
Diana, G. Bailey, T. (1997). Compounds, compositions and methods for treatment of hepatitis C. United States Patent No. 5,633,388.

**Peer**J ___________________________________________

## Supplemental Information

Supplemental information for this article can be found online at http://dx.doi.org/10.7717/peerj.74.

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
