# Peer review of "A holistic evolutionary and structural study of flaviviridae provides insights into the function and inhibition of HCV helicase"

_PeerJ, doi:10.7717/peerj.74_

## Round 0.1 · original submission · Minor Revisions

· Academic Editor

Minor Revisions

Both reviewers think the topic is interesting and the manuscript is well-written. I suggest the authors to fully address the issues raised by the reviewers. In particular, the main contribution should be clearly highlighted in the revision.

Reviewer 1 ·

Basic reporting

The paper is generally well written, and the structure and layout is appropriate for the material presented.

There are a few minor typos or omissions, eg p13 "Mangesium atom", p17 "SPC water(?)", such that a careful proof-reading is recommended.

Experimental design

In general the design of experiments seems entirely suitable, but a couple of relatively minor details to do with MD simulations caught my eye. None are fatal, but should be carefully checked and any inconsistencies ironed out.

p9 reports step size was set to 2 femtoseconds to a total of ten nanoseconds, but on p17 1 femtosecond to 50 ns is stated.

In addition, in NVT MD one would expect to fix volume of cell and let pressure vary, so statement on p9 "pressure at 1 atm" seems odd!

Validity of the findings

No comments

Additional comments

This is a high quality piece of research, using a range of computational methods to explore the biology of, and potential chemical interactions with, viral RNA helicases. The problem is clearly stated, the methods explained to a suitable level that would allow replication of results, and some interesting conclusions are reached.

·

Basic reporting

No Comments

Experimental design

The authors should improve their experimental design to make their results more clear.

Validity of the findings

The findings should be validated by other methods.

Additional comments

In this paper, Vlachakis et al. proposed an analysis of evolutionary and structure of flaviviridae. Some insights for the function and inhibition of HCV helicase have been provided in the results. The following lists my major concerns.

1. The main contribution of this paper should be clearly announced. The authors used some published bioinformatics methods and tools to reconstruct NS3 protein sequences, identify the conserved residues, and search the other features to dissect the HCV helicase. But the main biological insights or meanings of these identifications have not been clearly highlighted in the current version.

2. As for the usage of many existing bioinformatics tools, the version and the parameters in the implementation should be mentioned. It will be more convince to the readers that the identified patterns in the sequences and structures can also be detected by other similar methods. Otherwise, the presented results are highly biased only by one tool.

3. This writing of this manuscript is okay to follow, while it is not very clear about the motivation and the objectives when choose the methods to do which kind of analysis. The logic in the main steps and the motivation of each section should be improved.

4. The figures should be in higher resolutions. The acknowledgement section should be well prepared. In the current version, the reader will be confused about the funds for the research and that for the data source.

---

## Round 0.2 · accepted · Accept

· Academic Editor

Accept

The authors made significant improvement and sufficiently addressed all the concerns. I suggest its acceptance.